# Continual Learning of Neural Networks for Realtime Wireline Cable Position Inference

## Abstract

In the oil fields, Wireline cable is spooled onto a drum where computer vision techniques based on convolutional neural networks (CNNs) are applied to estimate the cable position in real time for automated spooling control. However, as new training data keeps arriving to continuously improve the network, the re-training procedure faces challenges. Online learning fashion with no memory to historical data leads to catastrophic forgetting. Meanwhile, saving all data will cause the disk space and training time to increase without bounds. In this paper, we proposed a method called the modified-REMIND (MREMIND) network. It is a replay-based continual learning method with longer memory to historical data and no memory overflow issues. Information of old data are kept for multiple iterations using a new dictionary update rule. Additionally, by dynamically partitioning the dataset, the method can be applied on devices with limited memory. In our experiments, we compared the proposed method with multiple state-of-the-art continual learning methods and the mREMIND network outperformed others both in accuracy and in disk space usage.

## 1 Introduction

With the emergence of AI-powered computer vision techniques in recent years, more and more vision-based applications are being introduced into the oil fields Wan (2019) Ant (2019). As an example, to automate the Wireline spooling process, one can mount a surveillance camera inside the cabin behind the window facing the drum, feed the live spooling video into a trained neural network and infer the so-called cable position in real time (Fig.1). This inferred cable position will then be passed to a spooling controller, which moves the spooling arm accordingly so that the cable can be laid onto the drum without gaps or crossovers Su et al. (2021b). To train the neural network, example video frames with ground truth cable position were first collected offline. The weights of the network were then learnt from these training frames to minimize the prediction error. After testing and optimization, the network was then deployed to the Wireline units. Training did not stop after the deployment, however, because the example frames in the first training process may not cover all the operational conditions (different Wireline units, drum sizes, cable types, lighting and weather conditions, operations in daytime vs. nighttime, etc). To continuously improve the network performance on the out-of-distribution frames, i.e., frames that look significantly different from those in the training set, we kept collecting new training videos to help generalize the performance of the network to various operational environments. This process is called continual, lifelong or incremental neural network training.

The most straightforward setup for continual learning is to combine the new videos with all the old ones and re-train the network using the ever-increasing number of training frames Su et al. (2021a). This, however, is not a scalable solution. Disk space required to hold all the past videos increases without bound over time. Also, the time to re-train a new network increases rapidly with more frames to extract, process and learn. A naive scalable solution could be to re-train the network only on the new videos and remove all the old ones. This solution, however, does not work well. During re-training, a neural network tends to forget what it learned previously, causing large prediction error when applying the updated network on the old videos. This phenomenon is called catastrophic forgetting in the literature Kirkpatrick et al. (2017). To help the network learn the characteristics of the new videos without forgetting the information of the old ones, continual learning techniques are usually applied.

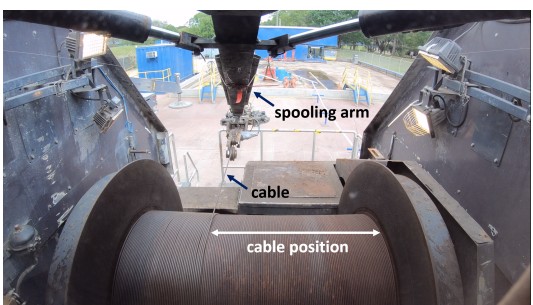

Figure 1: Wireline cable position is the distance between the contact point of the cable and the right flange of the drum. It is a dynamic variable needed by a spooling controller to control the spooling arm to automate the spooling process. We used computer vision to infer the cable position from a live spooling video in real time.

There exist three types of continual learning techniques Liu et al. (2020): the regularization-based method that penalizes large weight changes during re-training Kirkpatrick et al. (2017), the parameter isolation method that assigns different weights for different datasets Mallya & Lazebnik (2018), and the replay-based method that replays some of the old information to the network together with the new data Hayes et al. (2020) Wang et al. (2021) Prabhu et al. (2020). More descriptions of these methods will be provided in next section. One realization of the replay-based method is the REMIND network Hayes et al. (2020). Instead of saving all the raw videos, the REMIND network compresses and stores the features of the old videos and then replays them to the network for re-training. Feature compression is done by Product Quantization (PQ) Jegou et al. (2010), which clusters the features into K-means centroids and compactly represents them using the centroid indices. Although efficient in disk space usage, the original REMIND network replaces the previously stored centroids with those generated by the new data at each re-train iteration, which is not suitable for multiple re-training process. Besides, we found that the original REMIND network has memory overflow issue when the size of dataset is large.

In this paper, we proposed MREMIND, which modifies the dictionary update policy of the REMIND network so the previously stored centroids won't be replaced and deleted. To optimize memory allocation, we also partitioned the dataset into subsets, which enables the method to work on devices with limited RAM. Additionally, we investigated other state-of-the-art continual learning methods and compared their performance on the cable position inference problem. In summary, our contributions in this work are: 1. We applied continual learning techniques to train neural networks for real-time vision-based cable position inference to enable Wireline auto-spooling. 2. We modified the REMIND network with an extendable dictionary update policy and an improved memory management. 3. We compared the accuracy and the disk space usage of multiple state-of-the-art continual learning techniques on the cable position inference problem.

## 2 RELATED WORKS

Continual learning solutions aim to achieve the following goals: (1) Continuously train and re-train a network to learn from new data; (2) Keep the performance of the network on the previously seen old data; and (3) Avoid saving all the old data in its raw form to make the solution scalable for disk space usage. Related works in this area include:

- The regularization-based method. One example is the Elastic Weight Consolidation Method (EWC) Kirkpatrick et al. (2017). This method adds a regularization term to the loss function to penalize the change of the weights that are sensitive for the old datasets. Here "sensitivity" is quantified using the Fisher information. This method is simple to implement. However, setting and tuning the penalty stiffness can be highly problem-dependent. Also, the method was reported to be vulnerable to domain shift Wang et al. (2021) Aljundi et al. (2017).
- The the parameter isolation method. This method isolates the parameters and train different datasets on different parameters in the network Mallya & Lazebnik (2018). This method

indeed can solve the catastrophic forgetting problem because the previously learnt weights won't be affected by subsequent training. However, the amount of weights needed increases without bounds as more data becomes available. Therefore, it is not a scalable solution.

- The replay-based method. This method tackles the catastrophic forgetting problem by re-playing the old data during retraining. The old data may not in its raw form. It could be compressed, transformed, or reduced to a subset of the original data. For example, the REMIND network Hayes et al. (2020) extracted and stored features from the old data and replayed these features for network re-training. The ACAE-REMIND work Wang et al. (2021) increased the number of trainable parameters in plastic layer and introduced an auto-encoder to further enhance the inference accuracy. Wu et al. (2019) used generative adversarial network(GAN) to generate images of the old datasets for future replay. Interestingly, a recent paper, GDumb Prabhu et al. (2020), shows that by greedily storing the old images and then retraining the network from scratch at each iteration, one can get similar performance compared to other existing methods. For these replay-based methods, it is important to balance the distribution of replay samples for each dataset to avoid data bias.

## 3 Methods

In this section we propose a new continual learning method for wireline cable position inference task. In section 3.1, we will define the objective of the paper. In 3.2 we first give an introduction to current replay-based continual learning method and point out the drawbacks in actual deployment. Then we introduce our proposed MREMIND method.

### 3.1 Continual Learning For Wireline Automation

Upon actual deployment, wireline cable position inference might occur huge performance drop under different environments including lightness, weather and camera position setting. In this paper, our objective is to train a neural network that has high prediction accuracy in different configurations. Data under different settings will arrive sequentially. In order to avoid catastrophic forgetting issue as well as balance inference accuracy with training time and disk space usage, we apply continual learning technique to tackle this problem.

### 3.2 MREMIND: Modifed REMIND Network

#### 3.2.1 Current REMIND Network

Current REMIND method divides a neural network into two parts: (1) a feature extractor, also called the encoder or the frozen layer, and (2) a decoder, also called the plastic layer Hayes et al. (2020)Wang et al. (2021). The weights in the extractor are frozen after the first training, while those in the decoder are continuously trained on the new data iteration after iteration. In this paper, one "iteration" refers to one re-training of the network when a new dataset arrives. It could contain multiple training epochs. To avoid saving all the old raw videos, the REMIND network uses the frozen layer to extract the mid-level features from the old frames, then compresses and stores them into dictionaries. When a new dataset arrives, the old features are decompressed from the dictionaries and replayed to the network along with the new frames for network re-training. In this way, the network learns from the new data and keeps its performance on the old ones.

Training Procedure

To explain the training procedure of the REMIND network, denote *Enc* and *Dec* as the encoder and decoder of the network, respectively, Feeding an input image $x$ to the network produces a mid-level feature ($z = Enc(x)$) and a predicted cable position ($\hat{y} = Dec(Enc(x))$). Additionally, we use $y$ to denote the ground truth cable position, so that the prediction error is $\hat{y} - y$. The optimal weights of the network are trained by minimizing the squared prediction error. The training procedure is shown in Fig. 2 and is summarized in the steps below:

1. With all the weights in the encoder and decoder set to be tunable, train the network on all images in the first dataset 1. The trained network is the backbone pre-trained model.

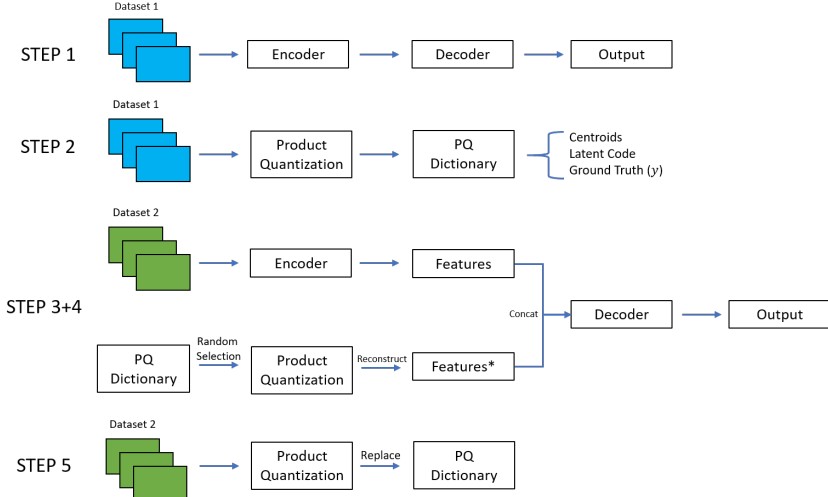

Figure 2: Illustration of the training procedure of the original REMIND network. At each iteration, old features are decompressed and replayed to the network along with the new features. The weights in the decoder is re-trained using the combined features by minimizing the prediction error. The new features are then compressed to replace the old ones.

2. Apply the PQ method to compress all the mid-level features from dataset 1. In this step, we obtain two base dictionaries representing dataset 1.

3. When a new dataset arrives, feed the new images into the encoder to extract features.

4. Uniformly and randomly select the compressed old features from the previous dataset, decompress and feed them together with the new features to the plastic layer. Re-train the decoder using these combined features by minimizing the prediction error.

5. Compress the new features, and replace the stored old features. In this step, we obtain two updated dictionaries representing the new features.

Product Quantization

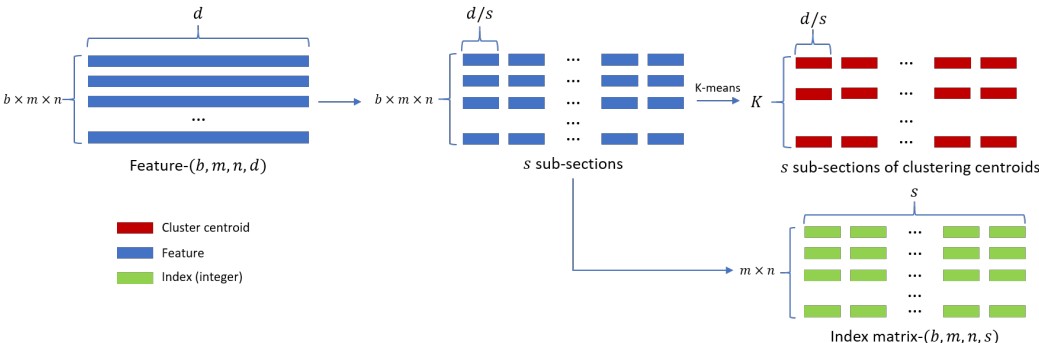

Figure 3: Illustration of the Product Quantization (PQ) method. A batch of features are first flattened into a 2D shape. Then the features are divided into sub-sections. K-means clustering is applied on each sub-section, resulting in K centroids and IDs of the centroids that represent the sub-features. The centroids and the centroid IDs will be used to recover the features during decompression.

The extracted features from the old frames are compressed and stored using the Product Quantization (PQ) method Jegou et al. (2010). The PQ method clusters the features into K-means centroids, and compactly represents the original features with the centroids indices. Each batch of features, with batch size $b$, number of channels $d$, and $m \times n$ pixels on each channel, can be flatten into a 2D matrix of size $(b \times m \times n, d)$. The PQ method divides the $d$ channels into $s$ sub-sections. K-means

clustering is then applied on each sub-section, resulting in K numbers of $d/s$-dimension centroids for each sub-section. The features can then be represented using the IDs of the closest centroids (of integer type). After compression, the original features are represented by the clustered centroids and the centroid IDs that tell us which centroid each sub-feature corresponds to. For example, image a batch of original features with the size of $b \times m \times n \times d \times 32$ bits (32 bits type). After PQ compression, we will get (1) an integer ID dictionary of the size $b \times m \times n \times \log_2 K \times s$ bits, and (2) a centroid dictionary of the size $K \times s \times (d/s) \times 32)$ bits (Fig.3). These two dictionaries are used to recover the features during decompression. It has been reported that the PQ method can store 50 times more features compared to raw images with the same disk space Wang et al. (2021). We used the library developed by Faiss for implementing the PQ method Johnson et al. (2017).

### 3.2.2 CHALLENGES

The two main drawbacks of current REMIND network Hayes et al. (2020) are: (1) The latent code dictionary update rule only allows one dataset being kept at the same time as old features will be replaced by new ones upon each arrival. So latent code only exist in the dictionary for 1 iteration. For example, when the third dataset arrives, features from the second dataset are gone and cannot be replayed to upate the parameters in the network. Thus, catastrophic forgetting can still occur in this case. (2) Memory overflow issue might occur upon the arrival of new dataset. Current REMIND model fits the complete dataset into the Product Quantization method for feature compression regardless of the size of the dataset. This will not work for computers with limited RAM and can lead to memory overflow.

### 3.2.3 DICTIONARY UPDATE RULE

Our proposed MREMIND method is implemented to tackle the issues that the original REMIND network is only capable of storing two datasets as well as memory overflow. For the dictionary update rule (Step 5) described in previous section, instead of replacing the dictionaries after each iteration, we compress the new features into new dictionaries and store them together with the old ones. An illustration of this modification is shown in Fig.4.

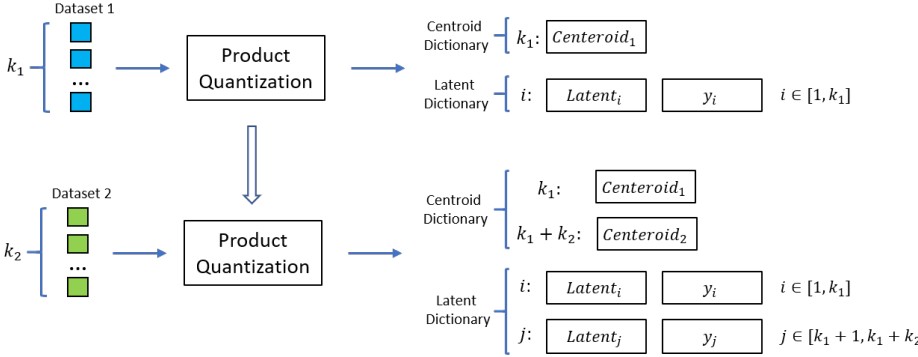

Figure 4: Illustration of the modified dictionary update rule. Instead of replacing the old dictionaries with the new ones, we keep both the old and new dictionaries in our modified REMIND network.

Each dataset is compressed to its own K-means centroids and indices. During the feature decompression process (the process of recovering the features from the centroids and the indices), one needs to know where to locate the centroids for the different datasets. To do this, we modify the data structure that stores the centroids, such that they are stored in a key-value pair structure: For each new dataset, we use the cumulative summation of the number of frames in all existing datasets as the key, and the centroids of the new features as the value. As an example, let's say we have 500 images in dataset 1, the key-value pair will be $\{500 : \text{centroid}_1\}$. Now imagine a second dataset arrives with 1000 images. We compress and obtain a new set of centroids for these 1000 images. At this point, we update the centroid dictionary to $\{500 : \text{centroid}_1, 1500 : \text{centroid}_2\}$. Here, the second key '1500' represents the cumulative summation of all the images (500 + 1000) from the two currently available datasets. On the other hand, the centroid indices and the ground truth cable positions are stored as: $\{i : (M_i, y_i), j : (M_j, y_j)\}, i \in [1, 500], j \in [501, 1500]$. During decompression, we use the

above data structure to reconstruct the features. For example, a feature with index 800 can be found to fall into the second centroid dictionary by comparing its index (800) to the keys (500 and 1500): $500 < 800 < 1500$. We note that the our modified dictionary update rule indeed requires a larger storage space because we save all the compressed features of the old datasets. However, because the floating-type features are converted into integer-type indices, the storage space increases at a much lower rate compared to other methods that store the raw images directly.

### 3.2.4 MEMORY OVERFLOW

For the memory overflow issue, current REMIND network applies the PQ method on the entire dataset regardless of the number of images in that dataset. For devices with limited RAM, this could lead to a memory overflow issue. To tackle this issue, we optimized the memory allocation policy by applying dataset partition. Denoting $D$ and $p$ as the size of the dataset and the designed maximum capacity of one subset, respectively. We divide a dataset into $D/p$ subsets and applied the PQ method on each of them individually. For example, if $D = 600, p = 256$, then the centroids will be stored as $\{256 : \text{centroid}_1, 512 : \text{centroid}_2, 600 : \text{centroid}_3\}$. The first key 256 represents the first partition with size $p = 256$, the second key 512 represents the sum of $p + p$. The last key 600 represents the final centroid matrix with $D - p - p = 88$ images. One can adjust the value of $p$ based on the hardware limits. It enables the REMIND network to work on devices with limited RAM.

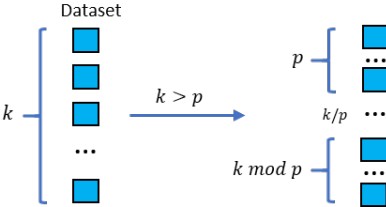

Figure 5: Illustration of dataset partition for enabling the REMIND network on RAM limited devices.

## 4 IMPLEMENTATION

### 4.1 DEVICES, DATASETS & DATA PREPROCESSING

We implemented all codes in Python 3.7 and PyTorch 1.5.1. All models are trained on a virtual machine in Google Cloud Platform using one Tesla K80 GPU (12 GB) with 16 GB RAM. The datasets in this paper were collected from Wireline trucks in Texas, USA. We used 3 datasets collected under different conditions to compare the continual learning solutions. Dataset 1 contained videos that were captured in daytime under sunny weather. Dataset 2 contained videos taken in nighttime in rains. Dataset 3 contained videos captured by a different camera with a few different camera settings: Dataset 1 & 2 had reflection protection so that the reflection from the window of the operators inside the cabin did not appear on the frames. For dataset 3, however, no reflection protection was set up. Besides, dataset 3 was taken under a special condition where the spooling arm never moved to near the left flange of the drum, causing a different distribution of the cable position compared to the other two datasets. For dataset 1 & 2, the cable position values ranged in between $[0, 1]$, but those in dataset 3 were in between $[0, 0.6]$. The ground truth cable position was calculated by another service during the Wireline operation. Fig.6 shows example frames from each of the 3 datasets. For training the REMIND network, we applied dataset 1 as our initial dataset to train the weights in both the frozen layer and the plastic layer. Then datasets 2 and 3 arrived and were used subsequently to re-train the plastic layer. All 3 videos were taken with 4K resolutions ($3840 \times 2180$) and a frame rate of 30.

We cropped the frames to show only the cable on the drum without the irrelevant background. During the training process, we resized the images to $(1200, 350)$ before feeding them into the network. After frame sampling, datasets 1, 2 and 3 contained 4838, 2007 and 3102 images, respectively. In terms of disk space, they used 4 GB, 1.5 GB and 5.6 GB, respectively. For all the methods being tested in this paper, we applied the ratio of 80%:10%:10% as the percentage of the

training/validation/testing sets. The evaluation metrics we applied to quantify the performance of network was the mean-squared error ($\mathcal{L} = (\hat{y} - y)^2$). The learning rate was $1e-4$.

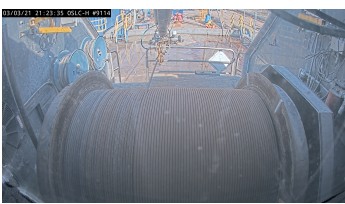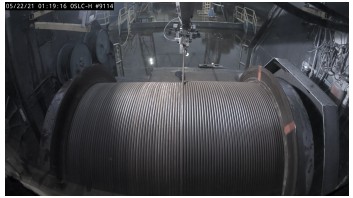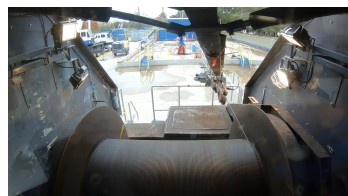

Figure 6: Example frames from datasets 1 (left), 2 (middle) and 3 (right). The environment (lighting conditions, weather, camera settings, frame distributions) under which the 3 datasets were taken were different. We used these 3 datasets to compare the performances of different continual learning methods.

## 4.2 THE BACKBONE NETWORK

We applied VGG19 Simonyan & Zisserman (2015) as the backbone network with the fully connected layers customized to output the Wireline cable position (a scalar).

## 4.3 BRUTE FORCE SOLUTION

In order to prove the existence of catastrophic forgetting in our datasets, we implemented the brute force solution (abbreviated as BF Sol) first, which followed a simple rule of 'train one drop one'. For this method, every time a new dataset arrived, the network was re-trained only on the new dataset without access to any old data.

## 4.4 STORE-ALL-TRAIN-ALL

For comparison purpose, we also implemented another naïve solution where all the old datasets were saved in its raw form on the disk (Store-All-Train-All, abbreviated as SATA). The combined training dataset increased without bounds in this method. At each iteration, the model was re-trained on this ever-increasing dataset, with the previously trained model as its initial condition.

## 4.5 THE GDUMB & GDUMB PLUS METHOD

For the GDumb method Prabhu et al. (2020), at each iteration, the images from all datasets were stored 'greedily' in the disk and the network was re-trained from scratch using the selected images. The GDumb method was shown to reach similar level of performance with the simplest configuration compared to all other methods. In addition to the original GDumb method, we also created an improved version and named it the GDumb Plus method. This new method greedily stored equal amount of images from each datasets (regardless of how many images the original dataset contained), and re-trained the network using the previously trained model as the initial condition (instead of training it from scratch).

## 4.6 THE ELASTIC WEIGHT CONSOLIDATION METHOD

For comparison purpose, we also implemented the Elastic Weight Consolidation (EWC) method Kirkpatrick et al. (2017). During re-training, a small subset of the previous dataset was forwarded into the model for a second time to update the Fisher Information matrix $\mathbf{F}$. A Fisher information dependent regularization term was added to the lost function for each weight. The regularization penalized the changes of those weights that were important for predicting the old data:

$$\mathcal{L} = \mathcal{L}_{\text{squared error}} + \sum_i \frac{\lambda}{2} F_i \left( \theta_i - \theta^*_{\text{prev},i} \right)^2 \tag{1}$$

Here $\mathcal{L}_{\text{squared error}}$ is the original squared prediction error loss term, $\lambda$ is the penalty stiffness, $F_i$ are the diagonal elements of the Fisher information matrix, and $\theta_i$ are the tunable weights in the network.

### 4.7 MREMIND NETWORK

Inspired by the work of ACAE-REMIND Wang et al. (2021), we tested separating the encoder and the decoder at different convolution layers when designing the modified REMIND network (abbreviated as MREMIND). For the first test, we used layer 1 to 29 of the VGG19 net as the frozen layer (encoder). This generated features of shape $(21, 75, 512)$. We named this model MREMIND-29. In the second test, we used layers 1 - 31 as the encoder. The features at layer 31 have the same shape as those at layer 29. We named this setting MREMIND-31.

## 5 EXPERIMENT RESULTS

We first trained the VGG19 network on dataset 1 with a batch size of 8 for 10 epochs to get the initial model. In this first training, the weights in both the encoder and the decoder were set to be tainable. Table.1 shows the prediction error when we applied this initial model (trained only on dataset 1) on all the 3 datasets. The table shows that a model trained on only one dataset does not perform well on the others. Therefore, continual learning solutions need to be applied.

| Method | Dataset 1 | Dataset 2 | Dataset 3 |
|---|---|---|---|
| Base Model | 1.02e-05 | 1.01e-03 | 3.02e-03 |

Table 1: Testing error on all the 3 datasets using the initial model that was trained only on dataset 1. This initial model did not perform well on datasets 2 and 3. Therefore, continual learning techniques need to be applied. Otherwise, the cable position inference wouldn't be accurate when the operational conditions change.

Next, we introduced dataset 2 into the continual training pipeline. We tested all the methods described in previous section. The number of data subsets used for MREMIND-29 and MREMIND-31 were 256 and 320, respectively. Size of code-book (ie., the number of sub-sections we divided the feature channels into for the PQ method) for MREMIND-29 and mREMIND-31 were 32 and 16, respectively. Comparisons of the testing errors and disk space usage were shown in Table.2 and Fig.7.

| Method | Dataset1 | Dataset2 | DUS$^+$ |
|---|---|---|---|
| BF Sol | 1.68e-03 | 3.34e-06 | 1500 |
| GDumb Plus | 4.58e-06 | 2.47e-06 | 3600* |
| GDumb | 1.19e-05 | 2.12e-05 | 3600* |
| SATA | 5.58e-06 | 3.79e-06 | 5400 |
| EWC-10000 | 9.70e-04 | 2.78e-06 | 1808 |
| EWC-1000000 | 9.19e-05 | 3.47e-06 | 1808 |
| MREMIND-31 | 7.89e-07 | 2.21e-06 | 1665 |
| **MREMIND-29** | **8.09e-07** | **2.10e-06** | **1809** |

$^+$ Disk space usage (DUS) in Megabyte (MB) does not include trained models
$^*$ Approximation usage

Table 2: Mean-squared prediction error, in units of squared meter, on datasets 1 & 2 after the network was trained for 2 iterations before dataset 3 was introduced.

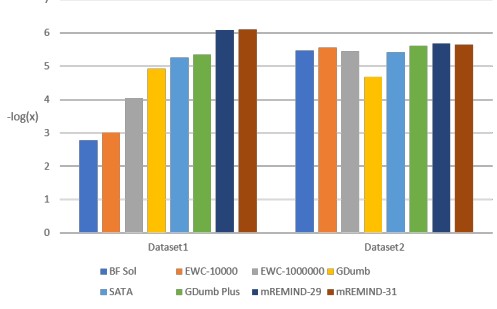

Figure 7: $-\log$ error plot on datasets 1 & 2 after the network was trained for 2 iterations.

It was shown in Table.1 that the testing error on dataset 1 using the initial model was 1.02e-05 $m^2$. At the end of the 2nd iteration, however, if one used the brute force method, the testing error on dataset 1 increased 168 times to 1.68e-03 $m^2$. This confirms that catastrophic forgetting occurred in our datasets and continual learning solutions are required to solve the problem. For the SATA (Store-All-Train-All) method, the accuracy was good, but it required a much longer training time and a significantly larger disk space usage. This is because all old frames were stored in their raw form. For the GDumb method, the original setting experienced no catastrophic forgetting on dataset 1. However, its performance on dataset 2 was not satisfying. The reason was that dataset 2 contained less images than dataset 1. When we uniformly sampled among both datasets, the network was retrained on more images from dataset 1 than from dataset 2. The GDumb Plus method solved this problem because it sampled equal amount of images from each dataset. For the EWC method,

because of the large domain shift from dataset 1 to 2 (see Fig.6), the method did not perform well and suffered large accuracy drop on dataset 1. This is in agreement with other studies Aljundi et al. (2017). Our modified REMIND network (MREMIND-29, MREMIND-31) achieved the overall best performance compared to the other methods in terms of accuracy and disk space usage.

Finally, we introduced dataset 3 into the re-training process. The number of data subsets and the code-book configurations for mREMIND-29 and mREMIND-31 were kept the same. The testing error and disk usage are shown in Table.3 and Fig.8.

| Method | Dataset 1 | Dataset 2 | Dataset 3 | DUS$^+$ |
|---|---|---|---|---|
| BF Sol | 4.93e-03 | 8.23e-03 | 1.82e-06 | 5600 |
| GDumb Plus | 5.79e-06 | 3.70e-06 | 1.92e-06 | 5550* |
| GDumb | 1.61e-05 | 2.62e-06 | 1.76e-06 | 5550* |
| SATA | 6.41e-06 | 3.82e-06 | 2.57e-06 | 11000 |
| EWC-10000 | 9.80e-04 | 2.00e-04 | 2.03e-06 | 6062 |
| EWC-1000000 | 5.50e-04 | 3.10e-04 | 1.92e-06 | 6062 |
| MREMIND-31 | 1.44e-06 | 6.21e-06 | 6.37e-07 | 5823 |
| **MREMIND-29** | **9.49e-07** | **2.55e-06** | **2.32e-06** | **6024** |

$^+$ Disk space in MB does not include trained models
$^*$ Approximation usage

Table 3: Mean-squared prediction error, in units of squared meter, on datasets 1 , 2 & 3 after the network was trained for 3 iterations. Disk space usage for each method is shown in the last column.

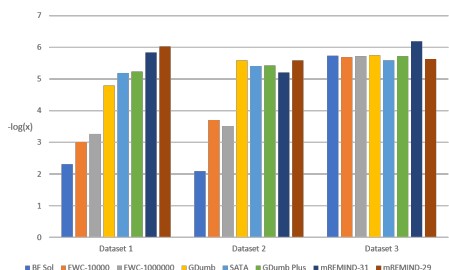

Figure 8: $-\log$ error plot on datasets 1, 2 & 3 after the network was trained for 3 iterations.

For brutal force solution, catastrophic forgetting became even worse. The testing error on dataset 1 increased from 1e-5 $m^2$ in Table. 1, to 1.68e-03 $m^2$ in Table.2, then to 4.93e-03 $m^2$ in Table.3 ($\sim$500 times larger compared to iteration 1, and $\sim$3 times larger compared to iteration 2). The GDumb method that trained the model from scratch at each iteration had larger testing error compared to the GDumb Plus method. As for the EWC method, tuning of the penalty stiffness was time consuming, yet the final performance was not satisfying. Our modified mREMIND-29 still had the lowest error in datasets 1 & 2, and best overall performance among all the 8 methods. In terms of disk space usage (DUS), as more datasets arrive, the GDumb-based and the brute force methods had relatively small space usages compared to SATA. Both the EWC and the modified REMIND methods stored higher-level representation of datasets. They were efficient in disk space usage.

## 6 CONCLUSIONS

Computer vision has the potential to help automate the Wireline spooling process. However, to develop a robust solution and to utilize the ever increasing and incrementally available field data, continual learning techniques are required. In this paper, we demonstrated that re-training a network only on the newly available videos can lead to catastrophic forgetting. On the other hand, saving and training on all historical videos is not a scalable nor a practical solution. We modified the replay-based REMIND network, and successfully applied it on the cable position inference problem. To compare its performance, we implemented a few other continual learning solutions, including the GDumb method, the GDumb Plus method, and the EWC method. The modified REMIND network gave the best performance among all these tested solutions. However, the GDumb Plus method is still an attractive solution because it is easy to implement with an acceptable performance.

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
