# OpenReview forum: "Continual Learning of Neural Networks for Realtime Wireline Cable Position Inference"
_ICLR.cc/2022/Conference — ICLR 2022 Submitted_

### Official Review · Reviewer_MCzH · 2021-10-30

**Correctness:** 2
**Technical Novelty And Significance:** 2
**Empirical Novelty And Significance:** 2
**Recommendation:** 3
**Confidence:** 4

**Main Review:**

### Weaknesses

#### Major
- This application is too narrow. The modifications are not coupled with this wireline auto-spooling task and can be applied on many other continual learning tasks (e.g. ImageNet ILSVRC-2012, CORe50, CLEVR, TDIUC tested in REMIND paper). But I am not sure if the proposed method works on those tasks. Commonly used benchmarks used to evaluate continual learning methods should be tested.

#### Minor
- Fig.2: Why do you use concat in "STEP 3+4" and how it works?
- Sec. 4.3: The network is re-trained from scratch? Better state more clear as in Sec. 4.4 and 4.5.
- Tab. 2: no comparison with REMIND?

**Summary Of The Paper:**

This submission modified REMIND with different dictionary update policy to keep track of previously stored centroids and partition the datasets to subsets to optimize memory usage. The proposed method has been verified on wireline auto-spooling (predict cable position). Several other state-of-the-art continual learning techniques have been tested on this task too.

**Summary Of The Review:**

Please refer to the Weaknesses

---

### Official Review · Reviewer_cXMu · 2021-11-02

**Correctness:** 4
**Technical Novelty And Significance:** 1
**Empirical Novelty And Significance:** 1
**Recommendation:** 3
**Confidence:** 5

**Main Review:**

* The paper is well written and easy to follow. The technical details are all clearly presented and technically sound.
* The contribution of this paper is very limited. The proposed method is an incremental work of REMIND network. Although the proposed two modifications may be effective, they are mainly ad-hoc tricks and lack enough insights and novelty.
* The proposed method is specifically designed for wireline cable position inference. Its impact to a larger scope may be limited. I suggest the authors to evaluate their approach to more diverse applications and compare against state-of-the-art continual learning methods on public benchmark datasets.

**Summary Of The Paper:**

This paper proposes a modified version of REMIND network for the specific problem of wireline cable position inference. It improves the original REMIND network by using an extendable dictionary update policy and optimizing the memory allocation policy through dataset partition, which addresses the catastrophic forgetting and memory overflow issue.

**Summary Of The Review:**

I think the contribution and novelty of the proposed method is somewhat limited and fails match the standard of ICLR.

---

### Official Review · Reviewer_kEWr · 2021-11-02

**Correctness:** 1
**Technical Novelty And Significance:** 1
**Empirical Novelty And Significance:** 1
**Recommendation:** 1
**Confidence:** 5

**Main Review:**

Strength
- The problem itself seems to be a good testbed for continual learning.

Weakness
1. The composition and details of the paper are quite unusual. For example,
- Most portion of the introduction section is dealing with the basic concept of continual learning instead of the issues that must be raised by the authors.
- A sentence or paragraph indented with bullet points should include important factors or summarization in a simplified manner. If the whole paragraph is to be indented in the 'related works' section, using bullet points would be inappropriate.
- I am pretty sure that the separation of subsections in section 4 is significantly weird. Backbone network can be mentioned in a single line in the implementation subsection and the following subsections are explaining too much about the methods that are not proposed by the authors of this paper.
- As MREMIND is built upon the work of REMIND, I understand that explanation of this method is necessary. But spending almost 2 pages out of 9 pages for the base method means that we cannot expect something new within those pages.
- The key points of Table 2 & 3 seem to be identical. Leaving only Table 3 would have been sufficient.

After all these factors, it has been hard to 'not' recognize that the authors had struggled to fill the paper with somewhat redundant information. I understand that offering prior information is as important as novel methods but honestly, this is too much.

2. In 3.2.2, when the third dataset arrives, shouldn't it be the first dataset that is forgotten? This sentence confuses my understanding. By the way, as a matter of fact, If I have understood right, REMIND does not abandon the old 'dataset' but the old 'sample' when the memory has reached its maximum capacity.  I want to hear a more specific explanation of this point from the authors.

3. The authors are saying that the memory overflow issue can be 'tackled' with the proposed method, but reducing the size of the data by dividing it into smaller subsets seems to simply 'delay' the overflow. If this is right, the contribution of this part (which is one of the two main contributions of this paper) is significantly minor.

4. If the authors are to claim that their method works better than any other previous methods, it is usually recommended to conduct comparison experiments on datasets and settings from them. It is hard to accept the experimental results performed on the author's custom datasets. Besides, putting aside that the evaluation metric is not well explained, prediction error seems to be seriously minor in all cases and the results must have fluctuated a lot. More justification of these results seems to be necessary.

5. If MREMIND is a method improving REMIND network, I wonder why the result of REMIND is not included in those two experiments.

**Summary Of The Paper:**

Original REMIND network compresses and stores features from old videos and replays them on the network at the time of retraining. Previously stored centroids are replaced with new ones and this is not suitable for a multiple re-training process. The authors proposed a new memory update rule to overcome catastrophic forgetting more efficiently and a strategy of resolving limited resource issues

**Summary Of The Review:**

The paper is quite below the standard of ICLR in all aspects of novelty, experiment, and writing.

---

### Official Review · Reviewer_jrD1 · 2021-11-03

**Correctness:** 3
**Technical Novelty And Significance:** 2
**Empirical Novelty And Significance:** 3
**Recommendation:** 5
**Confidence:** 4

**Details Of Ethics Concerns:**

No ethics concerns.

**Main Review:**

Strength:
- Well-written and easy-to-read paper.
- The motivation is clear where it has motivated the problem of forgetting and scalability.
- Experiment section contains figures and tables that show a proper measure of forgetting and memory disk usage.
Weakness:
- The challenge introduced in this paper is to overcome the current drawbacks of the REMIND network (Hayes et al. 2020). The latent code dictionary update and the memory overflow. However, there is no table or figure that shows how the proposed method performs against the REMIND network.
- The dataset used in this paper were collected from Wireline trucks in Texas, where the authors used 3 datasets collected under different conditions to compare the continual learning solutions. However, what happens with other datasets is unknown. So, I think the proposed approach should be evaluated with multiple datasets from different sources.
- Why MREMIND-29 performs better than MERIND-31 even though MERIND-31 has more samples (256 vs 320). Table 2, Table 3.
- How do you balance the distribution of replay samples to avoid data bias.


**Summary Of The Paper:**

In this paper, the authors proposed a method called the modified-REMIND (MREMIND) network. It is a replay-based continual learning method with a longer memory to historical data and no memory overflow issues. Information of old data is kept for multiple iterations using a new dictionary update rule. The modified-REMIND is applied to the cable position inference problem. experiment section shows that the modified-REMIND network gave the best performance among all other tested solutions.

**Summary Of The Review:**

Overall interesting paper. It explained the proposed idea very well. However, the experiment section is incomplete. I think the authors need to show how the proposed method perform using multiple datasets (i.e different than the Wireline trucks) such as (ImageNet, CORe50). Moreover, they need to compare against the REMIND network (Hayes et al. 2020), since the main contribution of this paper is to overcome the REMIND drawbacks.

---

### Decision · Program_Chairs · 2022-01-20

**Decision:**

Reject

**Comment:**

This submission receives four negative reviews. The raised issues include paper organizations, presentation clarity, more experimental evaluations, the trade-off between technical contribution and application configuration, and the potential impact on more general visual recognition scenarios. In the rebuttal and discussion phases, the authors do not make any response to these reviews. Overall, the AC agrees with four reviewers that the current submission does not reach the publication bar. The authors are suggested to improve the current submission based on the reviews to make further improvements.